# Lateral-Acceleration-Based Vehicle-Models-Blending for Automated Driving Controllers

**Jose A. Matute-Peaspan** [1,2,*], **Mauricio Marcano** [1,2], **Sergio Diaz** [1], **Asier Zubizarreta** [2] **and Joshue Perez** [1]

1   TECNALIA, Basque Research and Technology Alliance (BRTA), 48160 Derio, Spain; mauricio.marcano@tecnalia.com (M.M.); sergio.diaz@tecnalia.com (S.D.); joshue.perez@tecnalia.com (J.P.)
2   Department of Automatic Control and Systems Engineering, University of the Basque Country (UPV/EHU), 48013 Bilbao, Spain; asier.zubizarreta@ehu.eus
*   Correspondence: joseangel.matute@tecnalia.com

**Abstract:** Model-based trajectory tracking has become a widely used technique for automated driving system applications. A critical design decision is the proper selection of a vehicle model that achieves the best trade-off between real-time capability and robustness. Blending different types of vehicle models is a recent practice to increase the operating range of model-based trajectory tracking control applications. However, current approaches focus on the use of longitudinal speed as the blending parameter, with a formal procedure to tune and select its parameters still lacking. This work presents a novel approach based on lateral accelerations, along with a formal procedure and criteria to tune and select blending parameters, for its use on model-based predictive controllers for autonomous driving. An electric passenger bus traveling at different speeds over urban routes is proposed as a case study. Results demonstrate that the lateral acceleration, which is proportional to the lateral forces that differentiate kinematic and dynamic models, is a more appropriate model-switching enabler than the currently used longitudinal velocity. Moreover, the advanced procedure to define blending parameters is shown to be effective. Finally, a smooth blending method offers better tracking results versus sudden model switching ones and non-blending techniques.

**Keywords:** vehicle-model blending; trajectory tracking; model predictive control; automated driving; vehicle control

## 1. Introduction

Trajectory tracking is a crucial task in high driving automation developments. Thus, proper control methods must be designed to safely follow the desired reference path and speed. Model Predictive Control (MPC) is one of the most popular advanced model-based control techniques for this purpose. MPC approaches use the vehicle and tire models to predict the future behavior of the vehicle and compute the optimum control sequence to be applied. Moreover, this latter calculation is carried out considering explicitly the physical constraints of the system and its actuators [1].

A critical design decision in MPC-based methods is the selection of a proper model to predict the behavior of the vehicle, as the controller performance will depend heavily on its accuracy and real-time capability. However, this selection is not a trivial task, as covering a broad range of speeds, even the limits of handling, while maintaining real-time capabilities, represents a current engineering challenge [2].

In the literature, five vehicle-modeling techniques can be found: point-mass, geometric, kinematic, dynamic, and multi-body vehicle models [3,4]. The point-mass modeling considers the vehicle as a particle and it is commonly used in motion planning [5,6]. Even though it considers accelerations,

it ignores the turning capacity of the vehicle [4]. The geometric modeling considers the basic geometry of the vehicle and uses its geometric relationships for path tracking [7,8]. Although it offers good robustness in most low-speed maneuvers, it ignores the velocity and forces on the vehicle, which causes poor tracking performance at high speeds and transitional maneuvers [9]. The kinematic modeling is a simplified representation that, besides geometry, considers the orientation, velocity, and acceleration of the vehicle [10]. It provides appropriate performance at low speed (less than 5 m/s) when tire deformations are small and slips angles on the wheels can be neglected [11]. Experimental validations have shown a good performance of model-based controllers at low speeds [12,13]. However, when the lateral forces on tires increase (e.g., while turning at high speeds), its accuracy is compromised [14,15]. The dynamic modeling is a more complex vehicle representation that, besides geometry and kinematics, considers the internal forces and the inertia of the vehicle, providing accurate results in high-speed applications and extreme handling maneuvers [16–18]. Its implementation requires a tire model to estimate the longitudinal and lateral tire forces. For this purpose, a linear tire model is typically used, as it represents a good trade-off between computational efficiency and accuracy [19–21]. Finally, multi-body modeling is the most accurate representation of vehicle dynamics which is mainly employed as a virtual test platform for driving automation developments. Its high complexity and low computational efficiency make it difficult to implement this method today for real-time applications, therefore it is barely used for motion planning [22]. A comparison of these models in the context of this work is summarized in Table 1.

**Table 1.** Comparison of vehicle models in terms of performance and applications.

| Model | Strength(s) | Weakness(es) | Applications |
|---|---|---|---|
| Point-Mass | - Simplest model<br>- Easiest implementation | - Ignores minimum turning | - Motion planning |
| Geometric | - Considers minimum turn<br>- Robust in most maneuvers | - Ignores internal forces<br>- Ignores vehicle acceleration<br>- Not suitable for high speeds | - Motion planning/tracking<br>- Low speeds<br>- Constant speed/curvature |
| Kinematic | - Simple motion description<br>- Considers chassis slip | - No wheel's slip/skid<br>- Speed range is limited | - Motion planning/tracking<br>- Low speeds (<5 m/s)<br>- Varying speed/curvature |
| Dynamic | - Accurate motion estimate<br>- Handling dynamics<br>- Stability at handling limit | - Tire forces calculus<br>- Less numerical efficiency<br>- $\infty$ linear tire model ($V_x \approx 0$) | - Motion planning/tracking<br>- High speeds (>5 m/s)<br>- Varying speed/curvature<br>- Chassis slip angles < 5 deg |
| Multi-body | - Best accuracy<br>- All suspension forces | - Low numerical efficiency<br>- Complex implementation | - Motion planning<br>- Virtual test platform |

In MPC designs, kinematic and dynamic models are commonly used, as they present the best accuracy vs. computational cost ratio. Linear MPC approaches have been used to modify dynamically trajectory planners in the case of unexpected situations [23]. However, as seen in Table 1, each model presents strengths in different driving scenarios. Hence, in order to improve trajectory tracking and increase the speed range, recent works have proposed to combine these approaches [24]. This way, some authors have integrated both vehicle models in parallel to estimate more accurately relevant vehicle dynamics behavior, such as the side-slip angle [25] or the vehicle's position [26,27]. As the previous technique requires computing both models in parallel and increasing the computational effort, in recent years, the so-called model-blending approach has been proposed by some authors [28,29]. In this latter method, a model-switching strategy allows for selecting the most appropriate model depending on the driving scenario, allowing for increasing the validity range of the MPC-based vehicle control approach.

In the aforementioned works, to perform the model-blending technique, two aspects are usually considered: (1) the switching condition and (2) the switching method. The switching condition is the

criteria used to select the best model for each driving circumstance. This criterion has to be defined using a variable that allows for optimizing the operative range of each model so that the model-blending effectively improves trajectory tracking. In [28,29], the longitudinal speed is used as the switching condition, selecting the kinematic model to compute MPC predictions when the vehicle moves at low speed while using the dynamic model when moving at high-speed. However, as model validity is limited by the tire model saturation, it seems more appropriate to use another variable to perform the switching which is directly related to the tires' forces, such as lateral acceleration. Regarding the switching mode, previous works have presented two different methods—firstly, a sudden switch strategy that instantly changes between kinematic and dynamic models [28]; secondly, a progressive switching strategy that performs this change more smoothly using a linear approach [29]. Nevertheless, there is not a recommended procedure to blend models, besides the *trial-and-error* method.

In this work, a novel vehicle model blending procedure based on lateral acceleration for MPC is proposed. A coupled longitudinal and lateral vehicle dynamics is considered in the MPC formulations for trajectory tracking. The cornering stiffness is constantly estimated for a linear tire–road interaction. Lateral motion is constrained avoiding lane departures. The proposed approach presents three main contributions: (1) the lateral acceleration is selected as the switching condition instead of the longitudinal velocity, as the former is directly associated with the lateral forces on the tires and it defines a more consistent validity application range for each model; (2) a procedure proposal is presented to find the proper lateral acceleration switching value that allows for getting the best performance in terms of path tracking error; and (3) a thorough comparison of the proposed approach is carried out with the kinematic and dynamic vehicle models, as well as the longitudinal velocity as switching condition.

The structure of the paper is as follows: Section 2 presents the vehicle modeling and the combination method for kinematic and dynamic models; Section 3 defines the proposed model blending procedure tuning to achieve the best path-tracking in vehicle model blending; Section 4 details the case study and the MPC-based control architecture used to evaluate the model-blending approaches; Section 5 shows the results of the simulation experiments using a passenger bus in a realistic urban environment, with a detailed analysis and comparisons between switching methods; Section 6 closes with the conclusions and future works.

## 2. Vehicle Modeling

As previously stated, vehicle motion control based on MPC highly relies on accurate models. In addition, these need to be as simple as possible, so that the proposed trajectory tracking MPCs can be implemented in real-time platforms.

The single-track vehicle model is a well-known simplification broadly used in vehicle control approaches [11,30], where the front and rear wheels are defined as single wheels at each axle. The notation employed is depicted in Figure 1.

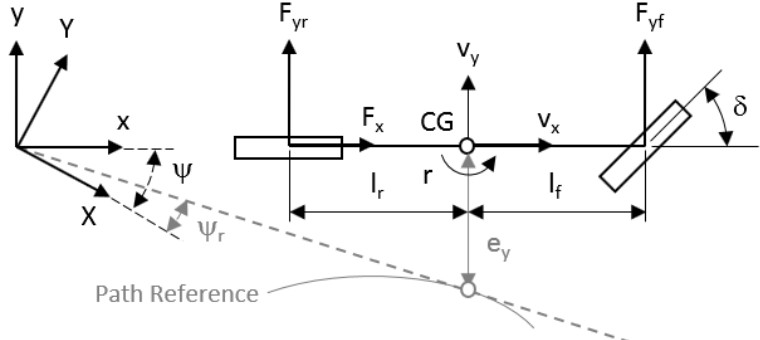

**Figure 1.** Vehicle modeling notation.

Based on this approach, kinematic and dynamic vehicle models are calculated. The equations of motion required for developing the trajectory-tracking control are detailed in Sections 2.1 and 2.2.

### 2.1. Kinematic Vehicle Model

Geometric relationships can be used to describe the motion of a vehicle under the assumption that lateral forces on tires do not considerably affect this estimation [11]. According to [31], the kinematic single-track vehicle model is even capable of generating feasible trajectories if the lateral acceleration ($\dot{v}_y$) does not exceed 0.5 g. The motion equations are given by Equation (1a–c):

$$\dot{v}_x = F_x/m \tag{1a}$$

$$\dot{v}_y = (F_x \tan \delta/m + v_x \Delta\delta/\cos^2 \delta)l_r/(l_f + l_r) \tag{1b}$$

$$\dot{r} = (F_x \tan \delta/m + v_x \Delta\delta/\cos^2 \delta)/(l_f + l_r) \tag{1c}$$

where $m$ is the total mass of the vehicle, $\delta$ is the front wheel steering angle, $\Delta\delta$ is the front wheel steering rate, and $l_f$ and $l_r$ are the distances from the center of gravity of the vehicle (CG) to the front and rear axles, respectively. The longitudinal velocity ($v_x$), lateral velocity ($v_y$), and yaw rate ($r$) are calculated with respect to the reference system attached to the CG. $v_y$ can be approximated to $v_y = l_r r$ considering $r = v_x \tan \delta/(l_f + l_r)$ [29]. The longitudinal force ($F_x$) acting on the vehicle is described as in Equation (2):

$$F_x = T_d P/r_{eff} - R_x - F_{aero} \tag{2}$$

where $T_d$ is the maximum acceleration and braking torque, $P$ is the throttle and brake pedals' positions and $r_{eff}$ is the effective tire radius. The force due to rolling resistance for radial-ply truck tires ($R_x$) is empirically described by [32] as in Equation (3):

$$R_x = (6e^{-3} \tanh v_x + 0.23e^{-6} v_x^2)mg \tag{3}$$

where $\tanh v_x$ is included to avoid numerical inconsistencies in MPC formulation when $v_x \approx 0$, $g$ is the gravity acceleration and $v_x$ is defined here in km/h.

Finally, the equivalent longitudinal aerodynamic drag force ($F_{aero}$) is defined by [11] as in Equation (4):

$$F_{aero} = 0.5\rho C_d A_f v_x^2 \tag{4}$$

where $\rho$ is the air density, $C_d$ is the drag coefficient, and $A_f$ is the frontal area.

### 2.2. Dynamic Vehicle Model

As the lateral force on the tires increases, the slip angles at the wheels are no longer considered negligible and a dynamic approach becomes necessary [11]. The motion equations in this approach are given by Equation (5a–c):

$$\dot{v}_x = (F_x - F_{yf} \sin \delta + mv_y r)/m \tag{5a}$$

$$\dot{v}_y = (F_{yf} \cos \delta + F_{yr} - mv_x r)/m \tag{5b}$$

$$\dot{r} = (l_f F_{yf} \cos \delta - l_r F_{yr})/I_z \tag{5c}$$

where $I_z$ is the yaw axis inertia. The external lateral forces on the front ($F_{yf}$) and rear ($F_{yr}$) axles are in Equation (6a,b):

$$F_{yf} = C_{\alpha f}\alpha_f \tag{6a}$$

$$F_{yr} = C_{\alpha r}\alpha_r \tag{6b}$$

where $C_{\alpha f}$ and $C_{\alpha r}$ are the cornering stiffness on the front and rear axles, respectively; and $\alpha_f$ and $\alpha_r$ are the slip angles associated with those axles, defined as

$$\alpha_f = \delta - \tan^{-1}((l_f r + v_y)/v_x) \tag{7a}$$
$$\alpha_r = \tan^{-1}((l_r r - v_y)/v_x) \tag{7b}$$

The estimation of $C_{\alpha f}$ and $C_{\alpha r}$ is a complex task in a real scenario, as these coefficients represent the interactions between tires and road surface, which may not be linear. Hence, in this work, a procedure to identify these parameters in real time is described in Section 4.2.

### 2.3. Blended Model for MPC Control

If the operating range of the MPC-based approach is to be increased, blending the aforementioned models is required. Thus, when lateral tire forces can be neglected, the kinematic model provides appropriate and fast results, while the dynamic model is used when tire slip is significant. This way, the set of nonlinear equations used for the longitudinal and lateral MPC is given by Equation (8a–g):

$$\dot{X} = v_x \cos \psi - v_y \sin \psi \tag{8a}$$
$$\dot{Y} = v_x \sin \psi + v_y \cos \psi \tag{8b}$$
$$\dot{\psi} = r \tag{8c}$$
$$\dot{\delta} = \Delta \delta \tag{8d}$$
$$\dot{v}_x = (1 - \lambda)\dot{v}_x^{kin} + \lambda \dot{v}_x^{dyn} \tag{8e}$$
$$\dot{v}_y = (1 - \lambda)\dot{v}_y^{kin} + \lambda \dot{v}_y^{dyn} \tag{8f}$$
$$\dot{r} = (1 - \lambda)\dot{r}^{kin} + \lambda \dot{r}^{dyn} \tag{8g}$$

where $[X, Y, \psi, \delta, v_x, v_y, r]^T$ are the states related to the CG. The positions $(X, Y)$ and orientation angle $(\psi)$ of the vehicle are considered in global coordinates, while the rest of the variables have been previously defined.

The control parameters $[\Delta \delta, P]^T$ are the front wheel angle rate and pedal positions, respectively. The use of incremental variables as $\Delta \delta$ (used in Equation (1b,c)) allows for including an integrative effect in the MPC controller, which is normalized for actuation stage consistency as detailed in Section 4.6.

The superscripts $(.)^{kin}$ and $(.)^{dyn}$ specify the relation of $v_x$, $v_y$ and $r$ with the kinematic and dynamic models defined in previous subsections. The parameter $\lambda$ is selected to switch or blend the two proposed vehicle models. If $\lambda = 0$, then a full kinematic model is applied. On the contrary, if $\lambda = 1$, then a fully dynamic model is engaged. An intermediate value of $\lambda$ defines a model blended circumstance.

The different strategies selected for model blending are detailed in depth in Section 4.3.

## 3. Tuning Procedure for Model Blending

As detailed in the Introduction, the blending of vehicle models is based on two aspects: (1) the switching condition, and (2) the switching method.

The switching condition is based on a physical measure usually available on the vehicle's acquisition, $v_x$ being the most used [28]. However, this value is typically defined by the designer by a rule of thumb based on several tests. A clear reference for this value is defined by [11] as 5 m/s, this being the recommended limit to employ the kinematic vehicle model. Nonetheless, this limit does not apply to all cases. For instance, in straight-line motion, lateral forces can be neglected, and the kinematic vehicle model could be considered valid in this condition even after 5 m/s.

The switching method is defined as how the switching condition occurs. As analyzed in Section 1, two main approaches have been proposed: a sudden or *step* change and a progressive one, in which a *linear*

blending is proposed. According to [29], a progressive transition between models offers a better response in the vehicle motion control in contrast to sudden switching conditions. However, the obtainment of $v_x$ values for this progressive blending is a complex task, as more than one reference for the switching condition is necessary and no more than *trial-and-error* methods are defined to achieve it.

In this work, a novel approach is proposed. The lateral acceleration $\dot{v}_y$ is considered as the switching condition parameter as it can be directly related to the current lateral forces on tires in any condition. The use of this variable is more consistent with the theoretical assumptions referred by [11] for the kinematic and dynamic vehicle models.

Based on this switching condition, the procedure proposed in Table 2 selects the best switching value of $\dot{v}_y$ for model blending in *step* and *linear* methods. The lateral ($e_y$) and angular ($e_\psi = \psi - \psi_r$) errors are considered as key metrics.

**Table 2.** Tuning procedure for model blending.

| Steps | Procedure |
|-------|-----------|
| **1.** | Plan a route for trajectory-tracking at constants $v_x^{ref}$ |
| **2.** | Execute motion control using kinematic vehicle model |
| **3.** | Execute motion control using dynamic vehicle model |
| **4.** | Average $e_y^{kin,dyn}$ values in a grid of $v_x^{ref}$ vs $\dot{v}_y$ |
| **5.** | Create surface plots from **4** |
| **6.** | For *step* switching method: |
| 6.a. | Make Linear Regressions (LR) of $e_y^{kin,dyn}$ vs $\dot{v}_y$ |
| 6.b. | Intersect LRs to find a $\dot{v}_y$ |
| 6.c. | The *step* switch is defined by **6b** |
| **7.** | For *linear* switching method: |
| 7.a. | Identify lowest $\dot{v}_y$ in surfaces intersection from **5** |
| 7.b. | Estimate difference between **6b** and **7a** |
| 7.c. | Use **6b** as point symmetry distance to **7a** |
| 7.d. | The *linear* switch is defined by **7c** |

This procedure will be applied for the case study proposed in the next section, and the results will be detailed in Section 5.1.

## 4. Case Study: An Urban Passenger Bus

In this section, the proposed vehicle model blending procedure is applied to a particular case study based on a trajectory tracking MPC for a passenger bus in an urban environment. The overall control architecture of the proposed case study, including the test vehicle and the MPC controller, is depicted in Figure 2.

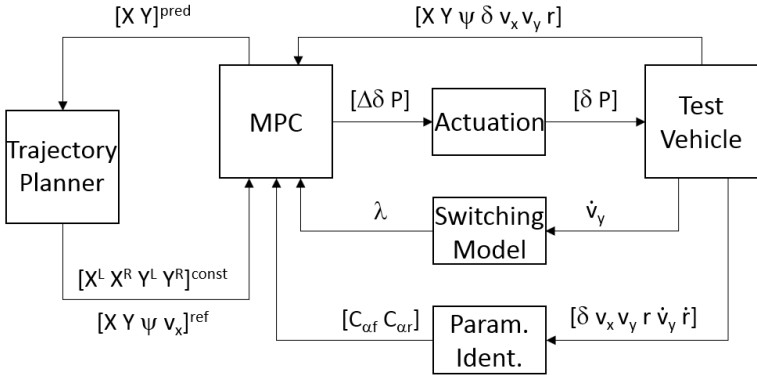

**Figure 2.** Control architecture.

### 4.1. Test Vehicle

The vehicle model employed is an electric passenger bus conceived for urban environments as depicted in Figure 3.

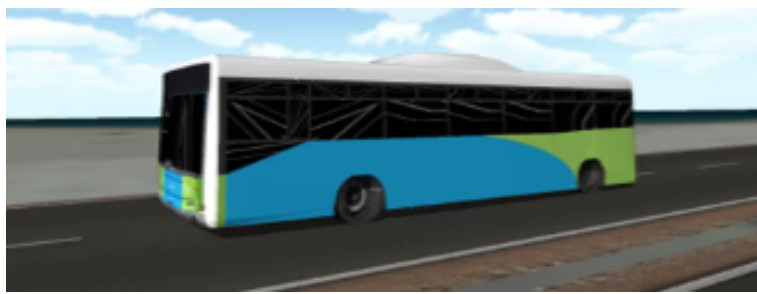

**Figure 3.** Test vehicle.

The test vehicle has been modeled in Dynacar [33] considering a multi-body formulation. A steering-knuckle-type suspension at the front axle and a rigid-axle-type suspension at the rear axle is linked to the chassis. Two wheels on the front and four wheels at the rear are linked to suspensions. The Pacejka tire model of a standard tire defined in [34] has been employed. Vehicle parameters are depicted in Table 3. Where $T_m$, $T_b$ and $T_r$ are the motor, braking, and regenerative brake torques, $N_t$ and $N_\delta$ are the transmission and the front-wheel-angle/steering-wheel-angle ratio, $I_t$ and $I_d$ are the transmission and drive-line inertia, respectively.

**Table 3.** Parameters of the Passenger Bus Model.

| Parameter | Value | Unit | Parameter | Value | Unit |
|-----------|-------|------|-----------|-------|------|
| $l_f$ | 3.55 | m | $m$ | 16,600 | kg |
| $l_r$ | 2.22 | m | $I_z$ | 115,063 | kg-m$^2$ |
| $A_f$ | 7.34 | m$^2$ | $C_d$ | 0.65 | - |
| $\rho$ | 1.21 | kg/m$^3$ | $g$ | 9.81 | m/s$^2$ |
| $T_m$ | 3600 | N-m | $N_t$ | 1:5.93 | - |
| $T_b$ | 12,000 | N-m | $N_\delta$ | 31:1 | - |
| $T_r$ | 35 | N-m | $I_t$ | 17 | kg-m$^2$ |
| $r_{eff}$ | 0.45 | m | $I_d$ | 100 | kg-m$^2$ |

### 4.2. Cornering Stiffness Identification

The dynamic model detailed in Section 2.2 requires the knowledge of the cornering stiffness coefficients that characterize the tire and road interaction. As the identification of these parameters is complex, an approach based on the *direct method* described by [35] is proposed.

One advantage of this method is the capacity to use the dynamic vehicle model employed for trajectory-tracking, as a simplified lateral tire model [11]. The $C_{\alpha f}$ and $C_{\alpha r}$ values depend on vehicle parameters as $m$, $I_z$, $l_f$ and $l_r$, and real-time measures of $\delta$, $v_x$, $v_y$ and $r$ as described in Equation (9).

$$\begin{bmatrix} C_{\alpha_f} \\ C_{\alpha_r} \end{bmatrix} = \begin{bmatrix} 2\alpha_f & -2\alpha_r \\ 2l_f\alpha_f & 2l_r\alpha_r \end{bmatrix}^{-1} \begin{bmatrix} m(v_y + v_x r) \\ I_z \dot{r} \end{bmatrix} \tag{9}$$

where $\alpha_f$ and $\alpha_r$ are defined previously in Equation (7a,b).

After the estimation of cornering stiffnesses, two separate one-dimensional Kalman filters reduce peak values from numerical inconsistencies in both $C_{\alpha f}$ and $C_{\alpha r}$. The filters are evaluated in the discrete-time domain, no control input is considered, and gain matrices related to states and measurement are constants valued as 1 [35,36]. The process and measurement noise covariances

are settled to 0.01 N/rad and 1 N/rad, respectively. In this work, the Kalman Filter block from MATLAB/Simulink was employed for this purpose.

In contrast to [35], the use of a linear Kalman filter avoids the definition of a threshold limit for $\alpha_f$ and $\alpha_r$, as the slip angles would approach to zero when the vehicle is driving straight or during transient steering maneuvers, affecting the cornering stiffness calculation. These results are explained in Section 5.3.

### 4.3. Switching Model

As previously stated, both the kinematic and dynamic models defined in Section 2 will be implemented in the MPC. Two different types of blending methods based on $v_y$ are proposed in this work for comparison purposes. Firstly, a *step* switch which causes a sudden change between kinematic and dynamic models. Secondly, a *linear* switch which executes a progressive change between models. The switching parameter $\lambda$ defined in Equation (a–g) is defined by Equation (10):

$$\lambda = \min[\max[\frac{|\dot{v}_y| - \dot{v}_y{}^{min}}{\dot{v}_y{}^{max} - \dot{v}_y{}^{min}}, 0], 1] \tag{10}$$

where $\dot{v}_y{}^{min}$ and $\dot{v}_y{}^{max}$ are the minimum and maximum acceleration thresholds defined by the switching designer (see Section 5.1 for more details). On the one hand, for the *step* method, $\dot{v}_y{}^{min} = \dot{v}_y{}^{max}$ is employed, where the change between *kin* and *dyn* models is performed when the sign from the estimation $(|\dot{v}_y| - \dot{v}_y{}^{min})/0$ results in $-\infty \vee \infty$, therefore switching $\lambda$ between $0 \vee 1$, respectively. On the other hand, for the *linear* method, $\dot{v}_y{}^{min} < \dot{v}_y{}^{max}$ is applied.

Additionally, a third switching strategy, called *speed* is considered for comparison purposes. This strategy, as proposed by [11], suddenly switches between the kinematic and dynamic models at 5 m/s and will be considered as a benchmarking strategy. Therefore, Equation (10) is also employed using $v_x$ as switching condition instead of $v_y$.

### 4.4. Trajectory Planner

The planned trajectory considers a realistic urban scenario with a total travel distance of approximately 680 m. It contains a couple of roundabouts with maximum curvatures ($k$) of around $0.08$ m$^{-1}$ connected through an avenue with smoother paths. The motion planner is based on parametric Bézier curves [37], considering the center-path of the road's right-lane. The starting vehicle's position and orientation, including the curvature segments, are depicted in Figure 4.

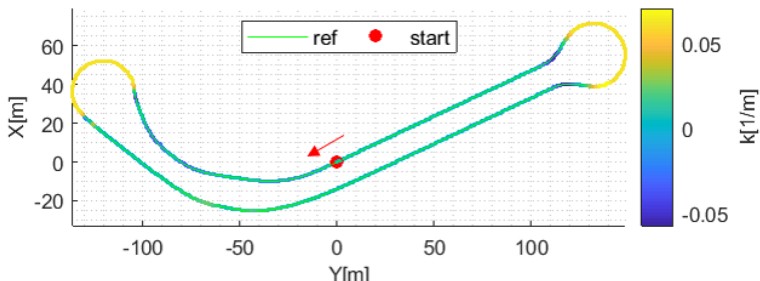

**Figure 4.** Planned trajectory considering the vehicle's maximum turning.

Considering the MPC's predictions ($[X, Y]$), new positions and orientations are estimated repeatedly at each iteration as new references for trajectory tracking ($[X, Y, \psi, v_x]$). Additionally, the center-lane path's border positions are continuously considered ($[X^L, X^R, Y^L, Y^R]$), using them as path constraints to avoid lane departures.

As this investigation is focused on the kinematic and dynamic models transitional effects over vehicle motion control, the round-about on the right side in Figure 4 is planned using non-smooth

curvatures provoking high lateral accelerations, which will help to analyze the model blending efficacy in extreme handling maneuvers.

*4.5. Model Predictive Control*

The developed MPC approach performs the longitudinal and lateral vehicle motion control of the trajectory defined in Section 4.4. It makes use of the blended model defined in Section 2.3, requiring an appropriate switching method for model change (i.e., the ones proposed in Section 4.3).

As the vehicle models are nonlinear, the proposed approach is a nonlinear MPC, which also includes a set of state and control constraints, designed to guarantee a safe execution of the dynamic driving task. The problem formulation is solved at each time step with a prediction horizon defined as $i, i+1, ..., i+N$ is presented in Equation (11a,d):

$$\min_{s(\cdot), u(\cdot)} \frac{1}{2} \sum_{i=0}^{N-1} \|s_i - s_i^{ref}\|_Q^2 + \|u_i\|_R^2 \tag{11a}$$

$$s.t.$$

$$s_{i+1} = f_d(s_i, u_i, \lambda_i), \quad i = 0, ..., N-1 \tag{11b}$$

$$\underline{s} \leq s_i \leq \bar{s}, \quad i = 0, ..., N-1 \tag{11c}$$

$$\underline{u} \leq u_i \leq \bar{u}, \quad i = 0, ..., N-1 \tag{11d}$$

where the $Q = \text{diag}(q_X, q_Y, q_\psi, q_{v_x})$ and $R = \text{diag}(q_{\Delta\delta}, q_P)$ are the weight matrices associated with the state tracking and control inputs, respectively (Equation (11a)).

The $s_{i+1} = f_d(s_i, u_i, \lambda_i)$ represents the blended model defined in Section 2.3. The states $s_i = [X, Y, \psi, v_x]_i^T$ are minimized according to the driving route geometry, orientation, and velocity references ($s_i^{ref}$). The control parameters $u_i = [\Delta\delta, P]_i^T$ are minimized to values as close to zero as possible to avoid sudden changes in control parameters. The switching parameter ($\lambda$) plays an important role in the formulation (Equation (11b)). The weights are set to $q_X = q_Y = q_\psi = q_{v_x} = 1$ and $q_{\Delta\delta} = q_P = 10$.

Constraints (Equation (11c,d)) are defined for both the states $s_i = [X, Y, \delta, v_x]_i^T$ as $[X^{L,R}, Y^{L,R}, \pm 0.68 \text{ rad}, v_x^{ref}]$; and control parameters $u_i = [\Delta\delta, P]_i^T$ as $\pm [0.5 \text{ rad/s}, 1]$. Keeping the vehicle on the planned path to avoid undesired lane departures is considered through additional soft constraints $[X^{L,R}, Y^{L,R}]$ as depicted in Figure 5. An additional distance ($d_w = 0.2$ m) is taken into account to avoid unfeasible solutions when results from $|X_i^L - X_i^R|$ or $|Y_i^L - Y_i^R|$ are near to zero.

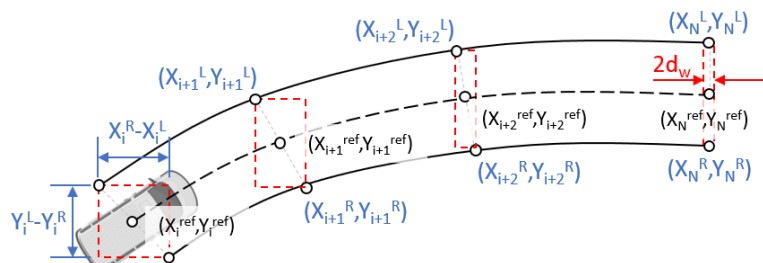

**Figure 5.** Path borders as constraints.

The path borders are obtained from the planned trajectory considering a continuous lane-width along the route, permitting a maximum lateral displacement from the center-lane path of 0.725 m. The constraint values for path borders are processed in real-time as described in Equation (12a–d):

$$\underline{X_i} = \min([X_i^L, X_i^R, X_i^{ref} - d_w]) \tag{12a}$$

$$\overline{X_i} = \max([X_i^L, X_i^R, X_i^{ref} + d_w]) \tag{12b}$$

$$\underline{Y_i} = \min([Y_i^L, Y_i^R, Y_i^{ref} - d_w]) \tag{12c}$$

$$\overline{Y_i} = \max([Y_i^L, Y_i^R, Y_i^{ref} + d_w]) \tag{12d}$$

Minimizing Equation 11a allows for calculating the optimum value of $u = [\Delta\delta, P]_i^T$ for the current time step. For that purpose, the nonlinear MPC is solved with the automatic code generator of the open-source ACADO toolkit [38], using QPOASES as the set solver, the sequential programming technique, and the direct multiple-shooting method for discretization. The prediction horizon is 5 s of look-ahead time considering a fixed time step among predictions of 0.5 s.

*4.6. Actuation Stage*

The control variable $\Delta\delta$ calculated by the MPC is integrated at this stage to obtain a $\delta$ normalized between $[-1, 1]$ considering a maximum value of $\delta = 0.68$ rad. The control variable $P$ is constrained between $[-1, 1]$ in the MPC formulation and represents the maximum brake and throttle pedal positions, respectively. Actuation delays of 150 ms for accelerator and 80 ms for both steering wheel and brake pedal were approximated by second-order transfer functions. In addition, rate limitations are applied mimicking a real actuation behavior [13,39].

## 5. Results and Discussion

The performance evaluation of vehicle models and switching methods employed are detailed in this section. The elements in the control architecture defined in Figure 2 and detailed in Section 4 are implemented in a MATLAB/Simulink setup which is used to perform a simulation-based analysis.

Considering the tuning procedure defined in Section 3, the parameters to perform the three switching methods introduced in Section 4.3 (*step*, *linear* and *speed*) are defined and evaluated first. In addition, pure kinematic (*kin*) and dynamic (*dyn*) methods are considered for comparison.

Three complete laps are simulated in the defined scenario (Figure 4), the results being recorded and evaluated. Eight values for $v_x^{ref}$ are defined from 1.1 m/s to 8.8 m/s, equally spaced at 1.1 m/s for each simulation test. This will allow for studying the influence of $v_x$ and the lateral acceleration ($a_y$) in the lateral motion control for the defined route.

*5.1. Tuning Procedure for Model Blending*

In this section, the procedure defined in Section 3 is applied to select the best switching value for $v_y$ for model blending in the *step* and *linear* methods. Note that the *speed* method is based on the $v_x$ as proposed in [11]. In the latter case, a *step* method is applied, in which a kinematic model is used below 5 m/s, and a dynamic model at higher speeds.

The results of the step-by-step procedure are detailed next.

**Steps 1 to 5:** Once the planned route for trajectory-tracking of Section 4.4 is defined, the vehicle motion control is executed using *kin* and *dyn* vehicle models at several $v_x^{ref}$ as described previously. The median is estimated for the absolute values $|e_y|$ and $|e_\psi|$ considering *kin* $(.)^{kin}$ and *dyn* $(.)^{dyn}$ models in a grid of $v_x^{ref}$ vs $a_y$. In practice, the median provides a better estimation in contrast to mean values for the *cut-off* definition pointed in *Steps 6a–c*. Results are processed through a two-dimensional convolution [40] creating surface plots as depicted in Figure 6.

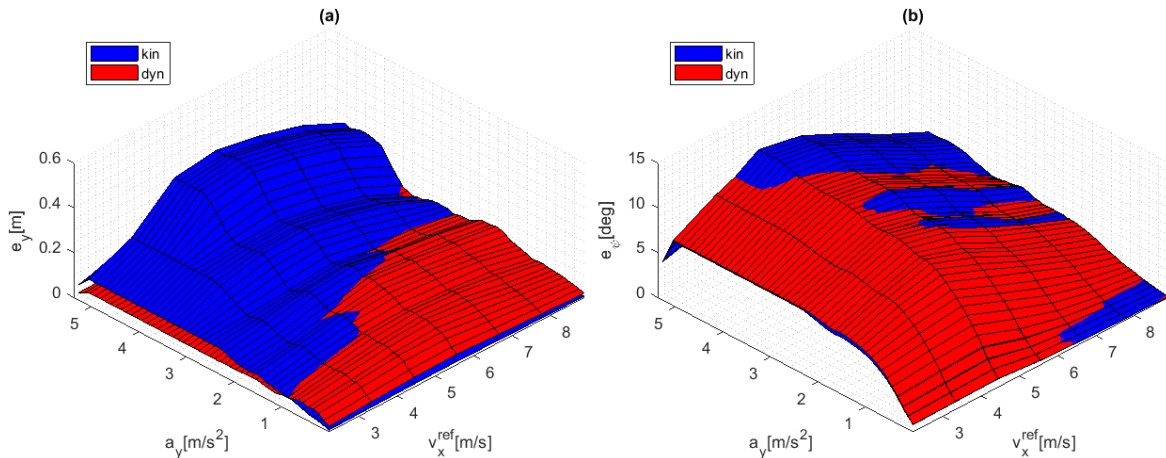

**Figure 6.** (**a**) $e_y$ and (**b**) $e_\psi$ for *kin* and *dyn* considering $v_x^{ref}$ and $a_y$.

The $e_y^{kin}$ and $e_y^{dyn}$ results are depicted in Figure 6a, in blue and red, respectively. The influence of $a_y$ is remarkable along several $v_x^{ref}$ tested, having a clear limit from *kin* and *dyn* surface intersections. These surface intersections help to prove the initial hypothesis which presents $a_y$ as a more appropriate switching condition than $v_x$.

The $e_\psi^{kin}$ and $e_\psi^{dyn}$ results are depicted in Figure 6b, in blue and red, respectively. There is no clear influence of $a_y$ or $v_x$ on the improvement of the path-tracking performance in terms of $e_\psi$, as *kin* and *dyn* models seem to have similar behavior. These findings motivate the idea of selecting $e_y$ over $e_\psi$ as the basis for a switching strategy.

**Steps 6a–c (*step* blending):** A linear regression is calculated from $e_y^{kin,dyn}$ vs $a_y$ as showed in Figure 7a (i.e., considering all the values of $e_y^{kin,dyn}$, associated with a certain $a_y$ and all related $v_x^{ref}$ values). The intersection of $LR^{kin}$ and $LR^{dyn}$ is approximately in 1.5 m/s$^2$, this being a useful cut-off value to define the switching condition to $a_y$. This value allows for obtaining the lowest $e_y$ values for *kin* and *dyn* models. As stated previously, if the same procedure is applied to $e_\psi^{kin,dyn}$ (Figure 7b), no relevant results can be extracted, as both models have similar performance.

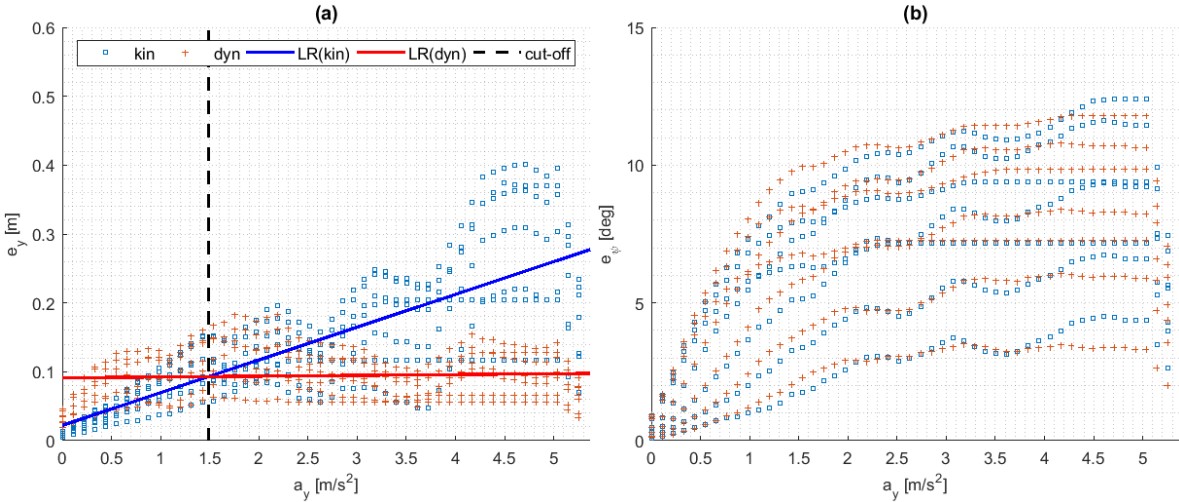

**Figure 7.** Linear regression in (**a**) $e_y$ finding the $a_y$ "cut-off" and (**b**) $e_\psi$.

**Steps 7a–d (*linear* blending):** The $a_y$ becomes relevant around 1 m/s$^2$ as depicted in Figure 6a. This is the lowest $a_y$ value that can be extracted from the surface intersection, which is useful for defining the initial condition of a progressive switching between models. In addition, the *step* switching (defined at 1.5 m/s$^2$) is considered as the point of symmetry to this initial condition. Therefore, the *linear*

switching is determined from 1 m/s$^2$ to 2 m/s$^2$ being centered around the *step* switching condition. The switching methods for model blending based on $\lambda \in [0, 1]$ are presented in Figure 8a.

*5.2. Trajectory-Tracking Response Analysis*

Results for three-of-eight simulation tests at constant $v_x^{ref}$ have been selected for discussion simplicity (2.2 m/s, 5.5 m/s, and 8.8 m/s). The *linear* method has been chosen for Figure 8b–d as it presents the best performance compared to other methods. The route values (black line) are located at zero values on *z*-axis as a reference, and the *z*-axis limits correspond to minimum and maximum estimation values of $v_x$, $a_y$, and $e_y$, respectively.

Figure 8b shows the $v_x$ of the bus for the *linear* method. Although the $v_x^{ref}$ is set as constant, note that the MPC regulates the final speed to avoid lane-departures (e.g., $v_x^{ref}$ = 8.8 m/s) as defined in Section 4.5. Hence, this is considered as a proper performance.

Figure 8c shows the $a_y$ of the bus. Larger values are obtained while turning as the $v_x^{ref}$ increases. Important transitions are observed mostly on the roundabout at the right-side due to non-smooth planned curvatures. This transitional behavior is observed in $a_y$ results independently of the tested $v_x^{ref}$, a phenomenon that is not acquired previously in $v_x$ results.

Figure 8d shows the $e_y$ of the bus, which is calculated by considering the road's center-lane and the current position at each time step. The transitional effects described in $a_y$ seem to affect the $e_y$ response, and, therefore, the path tracking.

The former results demonstrate that the MPC with the *linear* method provides an appropriate trajectory tracking.

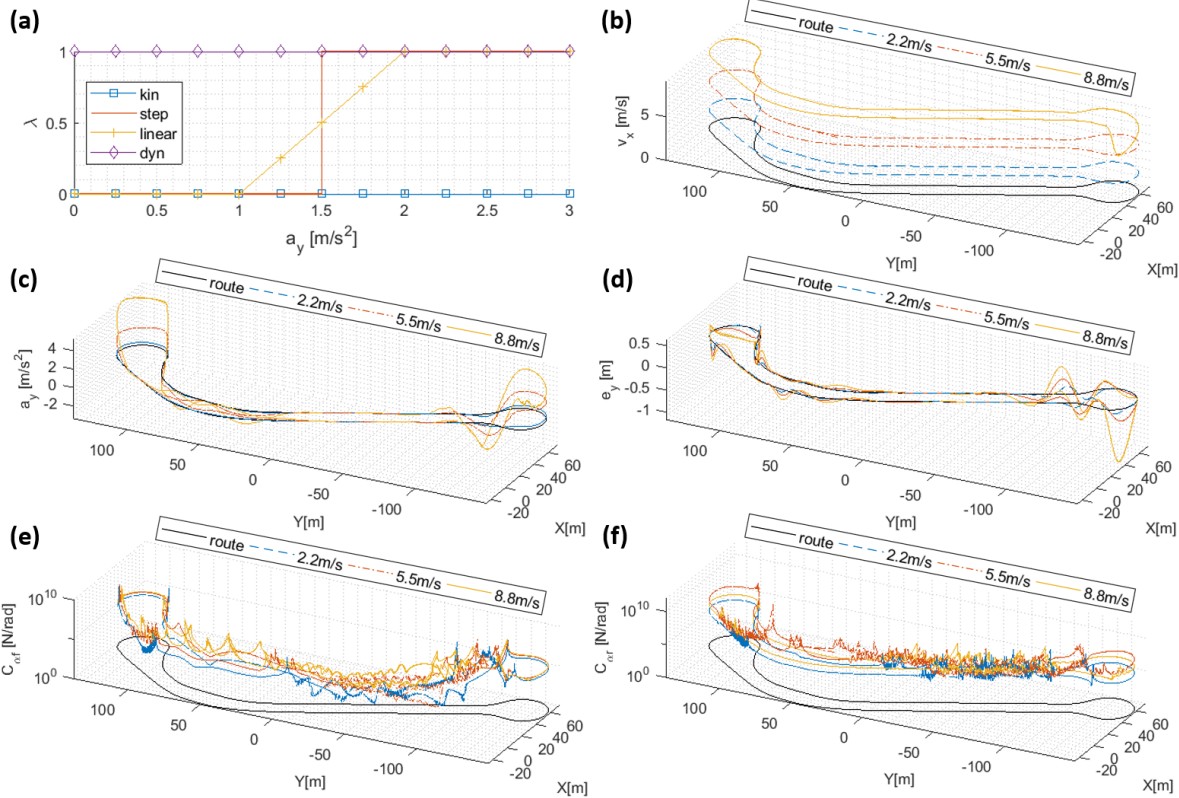

**Figure 8.** Results for: (**a**) switching methods for model blending; (**b**) $v_x$; (**c**) $a_y$ and (**d**) $e_y$ for *linear* method; (**e**) $C_{\alpha f}$ and (**f**) $C_{\alpha r}$ for the *dyn* method.

### 5.3. Cornering Stiffness Estimation Analysis

Results for three-of-eight simulation tests at constant $v_x^{ref}$ have been selected for discussion simplicity (1.1 m/s, 4.4 m/s and 8.8 m/s). The *dyn* method has been chosen for Figure 8e–f as it uses the cornering stiffness estimation in the whole range of $v_x^{ref}$. The route values (black line) is located at zero values on *z*-axis as a reference, and the *z*-axis limits correspond to minimum and maximum estimation values of both $C_{\alpha f}$ and $C_{\alpha f}$, respectively.

Results for $C_{\alpha f}$ and $C_{\alpha f}$ for the *dyn* method are presented in Figure 8e–f, respectively. These values are obtained at continuous turning maneuvers both at the front and rear axles. As expected, the estimations behave bumpily when steer angles are near zero, since they imply the inverse of a near-zero matrix (see Section 4.2). In practical terms, this results in larger variations of slip angles that are later attenuated by the linear Kalman filter. Consequently, the *dyn* method deteriorates the path-tracking at straight driving even if the vehicle is driving at high speed. This supports the main rationale to switch to *kin* method in this driving condition, even for $v_x > 5$ m/s, which is successfully achieved by the proposed $a_y$-based blending approach as opposed to the suggestion made by the existing $v_y$-based approach.

### 5.4. Lateral and Angular Error Analysis

Figure 9a,b shows the statistical distribution of $e_y$ and $e_\psi$ for five study conditions related to the five analyzed methods, allowing comparison for their trajectory-tracking performance. The boxes span (blue boxes) cover from 2% to 98% of the data, the whiskers span (black lines) cover from 1% to 99% of the data, the median (red horizontal lines) and mean ($\mu$, red plus signs) values as statistical metrics assessment.

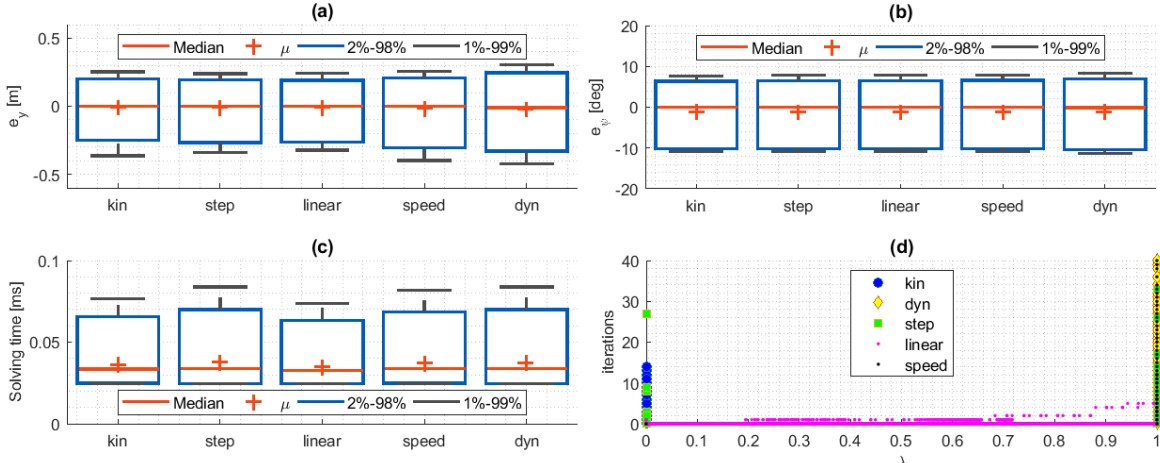

**Figure 9.** Results for: (**a**) $e_y$, (**b**) $e_\psi$, and (**c**) solving time statistics; (**d**) iterations number vs. $\lambda$ for different methods.

In Figure 9a, it can be seen that, for this low speed test track, the kinematic model (*kin*) clearly outperforms the dynamic model (*dyn*) as expected. However, blending these two models can produce better results than either one of them individually. Note that this particular track has the most low speed turns towards the left, whereas right-hand turns are mostly high speed. This allows for exemplifying the limitations of the *speed* method. Its positive $e_y$ distribution resembles the kinematic model behavior, while the negative side is much closer to the dynamic model. Since they can be correlated to the left and right-hand turns and thus the predominantly high and predominantly low speeds, it becomes clear that blending based on the speed uses either model in some cases where the other one behaves better (i.e., on the high speed turns, it uses the dynamic approximation even if the lateral forces are low and the kinematic model behaves better).

On the other hand, the hereby introduced switching strategies (*linear* and *step*), based on lateral acceleration, provide better behavior than the use of either kinematic or dynamic models, or even the aforementioned blending approach based on the speed (*speed*). This is achieved by actually switching when the lateral forces are significant, thus properly using the best approach in every condition to reduce errors (see Figure 6), rather than just avoiding singularities (which is the main motivation for the speed based blending). Since most of the track has low lateral acceleration, most of the error distribution for both $a_y$-based methods (*linear* and *step*) resembles the kinematic model behavior (see the blue boxes in Figure 9a, associated with the 2%–98% data interval). However, the black whiskers do show a significant improvement in reducing the lateral error corresponding to those cases with either high lateral acceleration and low speed or those of high speed and low lateral acceleration, thus proving the advantage of introducing the lateral acceleration as the blending parameter in place of the currently accepted vehicle velocity.

Furthermore, it is noted that the *linear* blending is slightly better than the *step* in terms of the maximum dispersion (black whiskers), though the actual advantage of this technique relates to the computational cost, as will become evident in the discussion below.

Figure 9b shows that the $e_\psi$ behaves very similarly regardless of the implementation of either of the analyzed methods, which fit the results shown in Section 5.1 and endorse the decision of considering $e_y$ surfaces for the blending procedure.

### 5.5. Computational Cost Analysis

To demonstrate the real-time capability of the presented approach, computational cost analysis has been carried out. The required time to calculate each control cycle of the proposed MPC controllers with the different blending methods has been evaluated. All controllers were execute at a 10 ms period on a LATITUDE E5570 provided with an Intel Core i7-6600U, CPU 2.60GHz (Santa Clara, California, USA). The results are depicted in Figure 9c, in which the statistical distribution of the solving time is depicted, following the same representation applied to Figure 9a–b. It can be seen where the worst-case scenario is for the *step* and *dyn* approaches, with mean values of 0.04 ms and maximums of nearly 0.08 ms. On the contrary, the *linear* method offers the best time efficiency with a mean value of 0.03 ms and a maximum solving time of 0.07 ms. Hence, results demonstrate that computational cost can be reduced by the use of blended models.

Note that all the referred approaches are based on a nonlinear MPC. In this case study, the previously calculated state and input values are used as a seed for the next iteration. Hence, when sudden or abrupt changes are required, the number of iterations required to solve the MPC problem increases significantly as depicted in Figure 9d. For instance, this happens when a sudden transition from a kinematic to a dynamic model is carried out in the *step* method. In this sense, the *linear* method reduces the required computational cost by lowering the number of iterations required to solve the optimization problem in the blending procedure to even slightly better values than the simple kinematic model.

### 6. Conclusions

The performance of MPC-based tracking controllers in automated vehicles is highly dependent on the selection of the model, either kinematic or dynamic. The kinematic enables accuracy at low $a_y$, e.g., driving straight. However, the dynamic provides better overall results when $a_y$ becomes representative, e.g., turning or steering transient maneuvers. To cover a wide operational range, switching or blending from one model to the other has been proposed in the literature. In particular, proposed approaches' use of the longitudinal velocity as the switching condition, which does not offer the best performance. Moreover, there is a lack of works related to the proper tuning of model-blending.

In this study, the use of the $a_y$ as opposed to the $v_x$ is proposed as the switching condition to blend vehicle models within an MPC-based trajectory tracking control. As tire forces are the critical factor for the validity of the kinematic/dynamic models, the $a_y$ is considered as a variable with direct relation to these forces, allowing for increasing the overall performance of the blended approach.

Additionally, a formal step-by-step tuning approach is proposed and detailed for two methods: *linear* and *step*.

The presented method is tested in a case study with an electric passenger bus in a virtual urban scenario. Results show that the proposed blending approaches based on $a_y$ provide a relative improvement of 15% in terms of $e_y$, in contrast to the method based on $v_x$ proposed in the literature. Additionally, it allows for reducing the maximum computational cost in 12% if a linear blending approach is used. Moreover, the validity of the tuning procedure is demonstrated.

Future works will assess the concepts presented in this research on both Hardware-in-the-Loop tests verification and real test platform validations, including the implementation issues related to parameters and variable estimations.

**Author Contributions:** Conceptualization, J.A.M.-P., and S.D.; methodology, J.A.M.-P., and S.D.; software, J.A.M.-P.; formal analysis, J.A.M.-P., and S.D., investigation J.A.M.-P. and S.D.; writing—original draft preparation, J.A.M.-P.; writing—review and editing, J.A.M.-P., M.M., S.D., A.Z., and J.P.; supervision, A.Z., and J.P. All authors have read and agreed to the published version of the manuscript.

**Funding:** This research was funded by AUTODRIVE within the Electronic Components and Systems for European Leadership Joint Undertaking (ECSEL JU) in collaboration with the European Union's H2020 Framework Program (H2020/2014-2020) and National Authorities, under Grant No. 737469.

**Conflicts of Interest:** The authors declare no conflict of interest.

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
