# Peer review of "Lateral-Acceleration-Based Vehicle-Models-Blending for Automated Driving Controllers"

_electronics, doi:10.3390/electronics9101674_

Round 1
Reviewer 1 Report
In this article, an approach based on lateral acceleration using blending parameters of vehicle models for automated driving controllers is proposed. Some simulations have been carried out using five methods proposed and the results were compared. This paper is well written and well organized. However, I have some additional comments:
It is not clear for me the functionality of equation (10), for the step method, since if the accelerations in "y", minimum and maximum, are equal, the value of lambda would be 1. Therefore, evaluating equations (8-e) and (8-g), it is noted that a dynamic model would be used. So, in this case, with the step model, the control would be only performed with the dynamic model and not the kinematic one, or with this model, Could It switch to either of the two controls? I consider that the explanation should be expanded because it can cause confusion.
It is only a format detail, the units must be separated from their magnitude. Please, check the paper format. On line 295, the unit is acceleration and not velocity. The variable is repeated on line 334, check this. In figures 8(d, f, g, h) do not show the legend of the curves, although there is a legend in Fig. 8(e), but it is not clear if this corresponds with the others.
Check if the values in figures 8 (b, c, i), and the respective text, are medians or means. Also, refer to the comments made in figure 6, to this respect.
It is not very clear to me that in figure 8b it is shown that ey is significantly affected by the implemented switching method since the three first blue boxes and its content are similar. I consider this explanation must be extended.
Author Response
Please see Reviewer 1 answers in the attachment

Reviewer 2 Report
The article contains only 3 keywords and this is not according to the Author’s Instructions.
The conclusions don’t contain a comparative analysis of the advantages of the proposed formal procedure to blend vehicle models within an MPC-based trajectory tracking control and those of existing models.To validate the proposed method (procedure), in addition to the simulation, experimental research is needed.
Author Response
Please see Reviewer 2 answers in the attachment

Reviewer 3 Report
The submitted paper attempts to solve a very complex problem of a combined modelling and model predictive control of a single-track vehicle. Lateral-acceleration and model-switching control are particularly concerned.
The paper is well-written and needs only minor English language corrections.
1) The reviewer is not sure whether the paper meets the aims & scope of Electronics journal. Maybe, it is acceptable within the Control & System Engineering.
2) Introduction section: The reviewer misses any link to the authors’ own works related to the problem in the state of the art section. This information helps to avoid suspicion of plagiarism.
3) Figure 1: The figure does not include “F_x” that is used in the following equations.
4) Why an electric passenger bus is used and the test vehicle, while all the models are valid for a single-track vehicle? Would it, e.g., a motorbike be a better test vehicle?
5) L. 105: The abbreviation “CG” is defined at l. 107, not here (with its first use).
6) L. 236: This line refers to “the discretized blended model defined in Sec. 2.3”. However, in Section 2.3, only the continuous-time model is provided to the reader. The reviewer believes that the discretized model should be explicitly given.
7) L. 258-259: What is the delay value?
8) The reviewer appreciates a detailed statistical analysis given in Section 5.
9) L. 379: One of “A.Z.” is superfluous.
If plagiarism is avoided, the paper can be accepted after fixing some minor issues raised above.
Author Response
Please see Reviewer 3 answers in the attachment

Reviewer 4 Report
This paper introduces a kind of model predictive controller (MPC) based on a hybrid vehicle model. This vehicle model hybrids the kinematic model and dynamic model. By measuring the lateral acceleration, this paper demonstrates a switching method to adjust the proportion of each vehicle model involved. The result shows that a smooth blending method gives a better path tracking. However, there are a few questions should be answered before proper publication.
- In section 4.2, linear Kalman filter is used to estimate cornering stiffness. Please explain how you use the LKF. The author uses tire slip angle in equation 9, but there is no information to get the tire slip angle.
 - In figure 8. (d) to (h), there are only kinematic, step, linear model results showing. There is no dynamic model result in the figure.
 - In figure 8. (g) to (h), please explain the reason for the change of lateral stiffness during straight travel and confirm the Z direction scale in the figure.

Author Response
Please see Reviewer 4 answers in the attachment

Round 2
Reviewer 1 Report
The responses of the authors to my comments have been appropriately performed. The new comments about figure 8 and the new figure 9 have improved suitably the explanation of the results obtained in the simulation.